# Effect of Scalp Cooling on the Pharmacokinetics of Paclitaxel

**DOI:** 10.3390/cancers13153915

**Published:** 2021-08-03

**Authors:** Leni van Doorn, Mandy M. van Rosmalen, Wendy M. van der Deure, Esther Oomen-de Hoop, Robert Porrazzo, Sophie M. Wijngaard, Ingrid A. Boere, Paola Veenstra, Eman Ibrahim, Peter de Bruijn, Lena E. Friberg, Stijn L. W. Koolen, Ron H. J. Mathijssen, Agnes Jager

**Affiliations:** 1Department of Medical Oncology, Erasmus MC Cancer Institute, 3015 GD Rotterdam, The Netherlands; m.vanrosmalen@erasmusmc.nl (M.M.v.R.); e.oomen-dehoop@erasmusmc.nl (E.O.-d.H.); r.porrazzo@erasmusmc.nl (R.P.); s.wijngaard@franciscus.nl (S.M.W.); i.boere@erasmusmc.nl (I.A.B.); p.veenstra@erasmusmc.nl (P.V.); p.debruijn@erasmusmc.nl (P.d.B.); s.koolen@erasmusmc.nl (S.L.W.K.); a.jager@erasmusmc.nl (A.J.); 2Department of Medical Oncology, Groene Hart Hospital, 2803 HH Gouda, The Netherlands; wendy.van.der.deure@ghz.nl; 3Department of Pharmacy, Uppsala University, 751 24 Uppsala, Sweden; eman.ibrahim@farmaci.uu.se (E.I.); lena.friberg@farmaci.uu.se (L.E.F.)

**Keywords:** chemotherapy-induced alopecia, scalp cooling, paclitaxel, clearance

## Abstract

**Simple Summary:**

This study investigated the correlation between scalp cooling used to prevent chemotherapy-induced alopecia and the pharmacokinetics of paclitaxel in female cancer patients with a solid tumor. In a prospective cohort study, 14 patients who were treated with weekly paclitaxel and scalp cooling were able to undergo pharmacokinetic sampling of paclitaxel during one cycle of treatment. In comparison to a control cohort of 24 patients treated with weekly paclitaxel without scalp cooling, our data showed that scalp cooling used concomitantly with one course of paclitaxel did not reduce or increase the clearance of paclitaxel. Therefore, it is unlikely that scalp cooling influences paclitaxel efficacy or toxicity. Finally, despite scalp cooling, half of the patients in our study developed a form of hair loss. Importantly, neither an association with difference in paclitaxel clearance nor change in hair loss was found.

**Abstract:**

Chemotherapy-induced alopecia (CIA), a side effect with high impact, can be prevented by cooling the scalp during the administration of some cytotoxic drugs. However, the effects of this prolonged scalp cooling on the pharmacokinetics of chemotherapy have never been investigated. In this study, we compared the pharmacokinetics of the widely used chemotherapeutic agent paclitaxel (weekly dose of 80–100 mg/m^2^) in female patients with solid tumors using concomitant scalp cooling (*n* = 14) or not (*n* = 24). Blood samples were collected in all patients for pharmacokinetic analyses up to 6 h after one course of paclitaxel administration. The primary endpoint was the clearance (L/h) of paclitaxel. Paclitaxel clearance—expressed as relative difference in geometric means—was 6.8% (90% CI: −16.7% to 4.4%) lower when paclitaxel was administered with concomitant scalp cooling versus paclitaxel infusions without scalp cooling. Within the subgroup of patients using scalp cooling, paclitaxel clearance was not statistically significantly different between patients with CIA (alopecia grade 1 or 2) and those without CIA. Hence, scalp cooling did not negatively influence the clearance of paclitaxel treatment.

## 1. Introduction

Chemotherapy-induced alopecia (CIA) is a commonly feared side effect of systemic anti-cancer treatment [1]. It can affect a patient’s quality of life dramatically and is one of the most distressing and adverse aspects of anti-cancer treatment, particularly for women [2]. Scalp cooling is a well-known method to try to prevent CIA during the administration of cytotoxic drugs for solid tumors [3,4]. Using scalp cooling, liquid refrigerant is pumped as coolant through a cooling cap that is placed on the head of the patient. In general, scalp cooling is started 20–45 min prior to, during, and up to 20–150 min after the chemotherapy infusion [5]. Scalp cooling results in a locally decreased blood flow due to vasoconstriction, resulting in a lower chemotherapy concentration at the root of the hair follicles and thereby, hopefully, in hair preservation [6]. The pharmacokinetics and pharmacodynamics of several drugs are influenced by body temperature [7]. Deep scalp cooling (4 °C), lasting for 20–45 min before, continued during and lasting for up to 150 min after chemotherapy infusion, may potentially lead to a temperature reduction of the whole body. This drop in body temperature may lead to alterations in pharmacokinetics [7]. This is of clinical relevance as changes in pharmacokinetics may lead to under- or over-exposure to the drug of interest. In a previous study, a physiologically based pharmacokinetic model (PBPK) of doxorubicin was modified to include a scalp skin compartment. The results of the model showed that maximum and average concentrations of doxorubicin in the scalp skin compartment were reduced by a factor of 3.6 and 1.6, respectively, during scalp cooling. These effects were due to reduced tissue perfusion and can positively influence the survival of hair follicles. However, mass transfer characteristics were not considered [8,9].

At present, there are no data available regarding the effects of scalp cooling on the pharmacokinetics of the cytotoxic drugs that are infused.

The severity of CIA, but also the success rate of scalp cooling, depends on the type of anti-cancer treatment used, its dose, method of administration and schedule of treatment [10,11]. Scalp cooling in patients treated with taxane-based chemotherapy such as paclitaxel, a widely used antineoplastic agent for the treatment of several cancers (e.g., breast, ovarian and esophageal cancer) [12,13,14], led to hair conservation in more than 50%, of patients, depending on the dose, compared with those who received no scalp cooling [15]. Scalp cooling is therefore offered as a part of standard treatment.

The aim of the present study was to investigate the impact of scalp cooling on paclitaxel pharmacokinetics in women who were scheduled to start treatment with paclitaxel and opted for scalp cooling compared to women who did not, of which there is a historical cohort. Although scalp cooling is usually a good option to prevent hair loss, it is unclear why some patients still develop CIA despite scalp cooling [2]. Therefore, we also studied the difference in paclitaxel pharmacokinetics between patients who developed alopecia compared to those who did not, despite scalp cooling.

## 2. Materials and Methods

### 2.1. Study Design and Patient Population

The aim of the study was to compare the pharmacokinetics of weekly paclitaxel between female cancer patients (aged over 18 years) who did use scalp cooling (SC+) concomitantly and who did not scalp cooling (SC−). The pharmacological data of the SC+ patients were prospectively collected (MEC-2015-140, date of approval 25 January 2016, Dutch Trial Registry; www.trialregister.nl (accessed on 2 August 2021; NL5543) and the pharmacokinetic data of the 24 SC− patients came from a previous single center pharmacokinetic study (MEC-2003-264, date of approval 19 February 2004, Dutch Trial Registry; www.trialregister.nl (accessed on 2 August 2021; NL2187). Both studies were performed at the Erasmus MC Cancer Institute, Department of Medical Oncology, Rotterdam, the Netherlands. All participating patients were asked to sign a written informed consent form. The studies were conducted according to the guidelines of the Declaration of Helsinki and approved by the Institutional Review Board and the local Ethics Committee of the Erasmus MC Rotterdam.

All patients were treated with paclitaxel infusions (combined with carboplatin) in a weekly dose of 80–100 mg/m^2^. Paclitaxel was dissolved in 250 mL sodium chloride 0.9% and infused in 1 h (i.e., an infusion rate of 250 mL/h). In those patients who had a history of hypersensitivity reaction to paclitaxel, a standard stepwise increase in infusion rates was used: 15 min at 5 mL/h, followed by 15 min at 12.5 mL/h and then continued at the normal infusion rate of 250 mL/h. Premedication (dexamethasone, ranitidine and clemastine) was administrated just before the paclitaxel infusion and in accordance with local standards.

Body temperature was only measured among the SC+ patients prior to the start of SC, at the start of the paclitaxel infusion (30 min after starting SC), 5 min prior to the end of the paclitaxel infusion and 180 min after ending the scalp cooling on three different body areas: in the mouth, in one ear and in one of the axillas. Scalp cooling was performed with a Paxman machine (PSC-2 Model, Paxman Coolers Limited, Huddersfield, UK) [16]. Hair was wetted before the start of scalp cooling. Cooling to 4 °C started 30 min prior to the infusion of paclitaxel and the cooling continued until 60 min after the administration of paclitaxel [10].

### 2.2. Pharmacokinetics of Paclitaxel

Blood samples for the pharmacological analyses were collected during one of the paclitaxel administrations, not necessarily the first administration, at four predefined time points: pre-dose, 55 (5 min prior to the end of the paclitaxel infusion), 90 and 360 min after the start of the paclitaxel infusion [17]. In patients where paclitaxel was administered in a standard stepwise increase in infusion rates because of hypersensitivity during previously administered paclitaxel treatment, this was pre-dose, 85 (5 min prior to the end of the paclitaxel infusion), 120 and 360 min after the start of paclitaxel infusion. The samples were collected by venipuncture or cannula in 4.5 mL lithium heparin blood collection tubes and processed within 10 min by centrifugation for 10 min at 2500–3000× *g* at 4 °C. Plasma was transferred into polypropylene tubes (1.8 mL Nunc Cryotube vials), which were stored at a temperature of minus 70 °C. Paclitaxel pharmacokinetics were measured in all plasma samples using a validated ultra-performance liquid chromatographic coupled to tandem mass spectrometry (UPLC-MS/MS) for precise quantification of paclitaxel plasma concentrations at the Laboratory of Translational Pharmacology of the Erasmus MC Rotterdam [18]. Non-linear mixed effects modeling was conducted using the software NONMEM [19]. A previously developed population pharmacokinetic model for paclitaxel [15], with two compartments describing the disposition and linear elimination, was used as a starting point. However, the model was here expanded to a three-compartment model to fit the data. Scalp cooling was tested for its effect on the model PK parameters. The model was used to obtain paclitaxel clearances and volume of distribution for each subject.

### 2.3. Chemotherapy-Induced Alopecia (CIA)

Patients received multiple courses of paclitaxel. Only patients without alopecia were eligible to participate in the study. The severity of alopecia was scored at the start of paclitaxel treatment and just before each treatment cycle thereafter by a physician or nurse practitioner according to the Common Terminology Criteria for Adverse Events (CTCAE) grades, version 4.03 [20]. CIA (according to CTCAEv4) is defined as grade 1 or 2 alopecia; grade 1 is defined as hair loss of <50% of an individual’s hair under normal conditions, not obvious from a distance but only upon close inspection, for which a different hairstyle may be required to cover the hair loss but a wig or hairpiece is not necessary; grade 2 is defined as hair loss of >50% of an individual’s hair under normal conditions that is apparent to others and for which a wig or hairpiece is necessary if the patient desires to camouflage the hair loss.

### 2.4. Statistical Analysis

The primary objective of our study was to investigate whether the clearance of paclitaxel was equivalent between patients who were treated with weekly paclitaxel with scalp cooling and without scalp cooling. The secondary objective was to determine the relation between paclitaxel clearance, temperature and CIA within the subgroup of patients using scalp cooling.

A sample size of 18 patients per group was required to demonstrate equivalence of SC+ and the SC−, with 80% power and a two-sided significance level of 0.05 and an original equivalence limit of one standard deviation (SD) based on the control group. However, since there were 24 controls available instead of the 18 patients required, the SC+ group could be reduced to 14 patients without a loss of power.

Based on advancing insight, it was decided to use the standard bioequivalence limits of 0.80 and 1.25 for the 90% CI of the ratio of the geometric means of paclitaxel clearance to draw conclusions about equivalence. Therefore, the analysis of clearance was performed on log-transformed values, as this parameter was assumed to follow a lognormal distribution [21]. Estimates for the mean difference in (log) plasma clearance and its 90% confidence interval (CI) were obtained by using the two-sample *t*-test. The mean difference and the 90% CI were exponentiated to provide the point estimate of the geometric mean ratio and the 90% CI for this ratio that can be interpreted as a relative difference (RD) in percentages by using the following equation: RD = (geometric mean ratio − 1) × 100%. The difference in paclitaxel clearance between patients who developed hair loss versus those who did not was analyzed similarly to the analysis of clearance. However, as the aim here was to study whether there was a difference, the 95% CI was used for this analysis. The differences in temperature over time were analyzed by location by means of a mixed model with a random effect for each patient.

Patient characteristics from the prospective study and the control cohort were presented as medians and interquartile ranges (IQR) or as numbers with percentages. Clearance was described per study and hair loss group by means of the geometric mean and the coefficient of variation (CV). All statistical analyses were performed using Stata version 16.1 (StataCorp, College Station, TX, USA).

## 3. Results

### 3.1. Patients Characteristics

Between January 2016 and December 2020, a total of 21 female patients with solid tumors were enrolled in the scalp cooling study. Seven patients were excluded due to incomplete blood sampling. Hence, 14 patients were evaluable for the main analyses. These patients were treated in accordance with the study protocol with scalp cooling during one cycle of paclitaxel treatment dosed at 80 mg/m^2^ (36%) or 90 mg/m^2^ (64%) depending on the indication. Three patients were treated with the standard stepwise increase in infusion rates because of hypersensitivity during previously administered paclitaxel treatment. The median paclitaxel infusion duration was 1.08 h. The post infusion cooling time was 60 min for all patients after which the scalp cooling was removed. Baseline demographic and clinical characteristics of the SC+ group and the SC− group are summarized in Table 1.

### 3.2. Effect of Scalp Cooling on Paclitaxel Pharmacokinetics

Paclitaxel concentrations of the pharmacokinetic profile of patients using scalp cooling (*n* = 14) and patients not using scalp cooling (*n* = 24) are shown in Figure 1.

Pharmacokinetic results of 14 SC+ patients and 24 SC− patients are depicted in Table 2. Paclitaxel clearance was 6.8% (90% CI–16.7% to 4.4%) lower when paclitaxel was administered with concomitant scalp cooling compared to paclitaxel administration without scalp cooling. The distribution of paclitaxel was 5.9% (90% CI–2.3% to 14.8%) higher with concomitant scalp cooling compared to paclitaxel administration without scalp cooling. The scalp cooling did not have a significant effect on the model PK parameters.

### 3.3. Temperature Course during Scalp Cooling

The course of the temperature during the scalp cooling of 14 patients, measured on three different locations, is shown in Figure 2. No significant difference in time could be observed in the mouth (overall *p*-value = 0.238). For both ear and axilla, a significant difference was found between baseline and the 30 min point (*p* = 0.001 and *p* = 0.039, respectively). Furthermore, a difference was also found between baseline and the 55 min point in the ear (*p* = 0.003).

### 3.4. Chemotherapy-Induced Alopecia (CIA) during Scalp Cooling

Despite adequate scalp cooling during paclitaxel treatment, seven patients reported CIA during scalp cooling grade 1 (*n* = 4) and grade 2 (*n* = 3), whereas the other seven patients reported no CIA during scalp cooling. There was no difference in the median dosage of paclitaxel. More patients with than without CIA, despite scalp cooling, had a decrease of >1 °C in at least one of the measurement sites (see Table 3).

### 3.5. Paclitaxel Clearance with or without CIA after Scalp Cooling

Within the SC+ group, paclitaxel clearance and Vd was not significantly statistically different between patients with CIA (alopecia grade 1 or 2) and those without CIA (Table 4).

## 4. Discussion

In the present study, we demonstrated that clearance as measure for systemic exposure and the volume of distribution of paclitaxel did not reduce or increase as a result of scalp cooling. Although the decrease in body temperature during scalp cooling at each site measured was small, the decrease was statistically significant different during scalp cooling for the measurement in the ear and axilla. There was no significant difference in temperature until 3 h after scalp cooling was discontinued. Finally, half of the patients developed some form of hair loss despite scalp cooling. However, this was not associated with paclitaxel clearance.

Mild hypothermia (body cooling to 32 to 34 °C for 12 to 48 h) can alter the pharmacokinetic parameters of several drugs [7]. Although the mechanism(s) behind changes in drug levels due to hypothermia has not been fully elucidated, impaired hepatic metabolism is likely, possibly via its effect on cytochrome P450 metabolism. In a study in healthy volunteers, for example, the clearance of midazolam as an index of CYP3A4/5 metabolism decreased by 11% for every degree Celsius decrease in a core temperature of 36.5 °C [22].

In our analysis, a population PK model consisting of a central compartment with two peripheral compartments connecting to it was used to obtain clearances and volumes of distribution of each subject. Unlike PBPK models, these compartments have no anatomic or physiological significance. However, they can still be used to investigate the influence of subject characteristics (i.e., scalp cooling) on the predicted subject PK parameters describing the whole-body drug disposition [23]. Considering the scalp skin drug disposition in particular would require more data, and it is outside the scope of our study.

To the best of our knowledge, this is the first study investigating the influence of scalp cooling on the pharmacokinetic outcome of anti-cancer drugs. We hypothesized that the decrease in body temperature as a result of scalp cooling may influence the pharmacokinetics of paclitaxel. Although a small decrease in body temperature was found after 50 min of scalp cooling, which did not fully return to baseline after 5 h, the absolute decrease was limited: 60% of the women had a temperature drop of less than just 1 °C. This drop in temperature may be too small to demonstrate a difference in clearance of paclitaxel.

Scalp cooling is usually a good option to prevent hair loss for anti-cancer drugs with a short half-life or a rapid systemic distribution, such as paclitaxel [11]. However, it is unclear why some patients still develop CIA despite scalp cooling during such chemotherapy administration. In our study, half of the women were found to have some form of hair loss, of whom more than 40% developed grade 2 alopecia (although no one developed full baldness). This is somewhat higher than mentioned in previous studies, probably due to differences in definition of CIA between these studies [11]. We found no clear difference between the patients who developed CIA and those who did not. If any, the mean temperatures were slightly lower among those women without CIA compared to those with CIA. It is important to emphasize that CIA was also not due to differences in paclitaxel clearance. Further research is needed to identify possible explanations to better advise future patients about the chance of hair preservation with scalp cooling.

Some potential shortcomings of our study need to be mentioned. To answer our research question, the pharmacokinetic data from the controls (SC−) were collected prospectively as a part of a separate study. However, the method used to calculate paclitaxel clearance was similar in both studies, allowing for the pooling of results. Since patients had received paclitaxel at different doses (range 80–100 mg/m^2^), the clearance of paclitaxel was used as the primary endpoint, as this result is unaffected by dose differences. Although the dose of paclitaxel strongly determines the success rate of scalp cooling for hair preservation, within the dose range of 80–90 mg/m^2^, this had no effect on CIA in our study. In three patients, the total infusion time of paclitaxel was somewhat longer than the standard 60 min infusion time due to a history of hypersensitivity reaction to paclitaxel. This also may have a slight effect on the pharmacokinetics. However, infusion time was taken into consideration when calculating the clearance for each patient.

## 5. Conclusions

Our data showed that scalp cooling concomitant with paclitaxel did not reduce nor increase the clearance of paclitaxel. Therefore, it is unlikely that scalp cooling influences paclitaxel efficacy.

Finally, despite scalp cooling, half of the patients in our study developed a form of hair loss. Importantly, neither an association with difference in paclitaxel clearance nor a change in hair loss was found. Further research is warranted to optimize hair preservation in patients treated with paclitaxel.

## Figures and Tables

**Figure 1 cancers-13-03915-f001:**
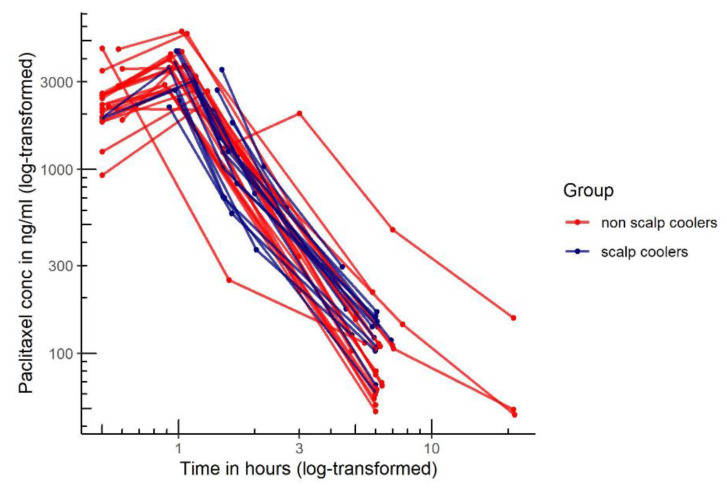
Paclitaxel concentration time profiles of the pharmacokinetic profile of patients using scalp cooling (*n* = 14, blue line) and patients not using scalp cooling (*n* = 24, red line).

**Figure 2 cancers-13-03915-f002:**
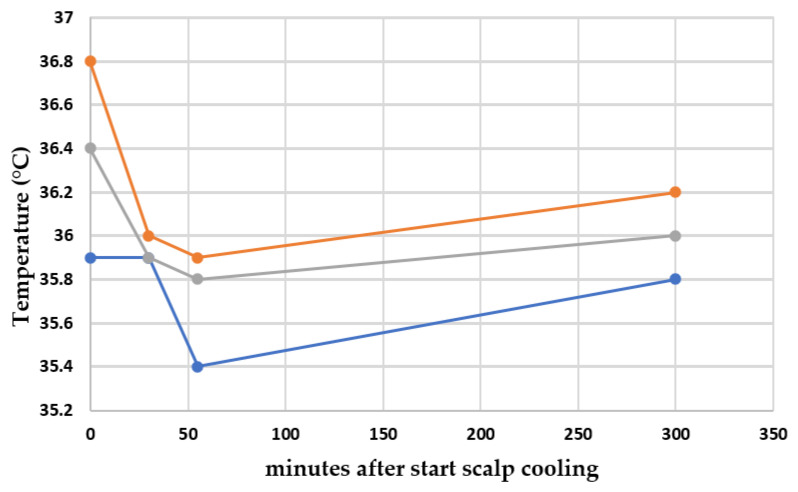
Course of median temperature during the scalp cooling of 14 patients in an ear, the mouth and an axilla. Locations of temperature measurements: orange line: an ear; gray line: the mouth; and blue line: an axilla.

**Table 1 cancers-13-03915-t001:** Baseline characteristics.

Baseline Characteristics	Scalp Cooling (SC+) (*n* = 14)	No Scalp Cooling (SC−) (*n* = 24)
Female, *n* (%)	14	24
Age (years) median [IQR]	51 [46–62]	61 [54–65]
Paclitaxel treatment dose, *n* (%)		
80 mg/m^2^	5 (36)	10 (42)
90 mg/m^2^	9 (64)	8 (37)
100 mg/m^2^	0	4 (21)
Paclitaxel dose (mg), median [IQR]	150 [143–160]	153 [143–170]
Infusion time (h), median [IQR]	1.08 [1.00–1.32]	1.01 [0.92–1.58]
BSA (m^2^) median [IQR}	1.71 [1.63–1.87]	1.80 [1.63–1.93]
Indication, *n* (%)		
Breast cancer	5 (36)	11 (46)
Cervix cancer	9 (64)	5 (21)
Esophageal cancer	0	3 (12)
Ovarian cancer	0	5 (21)

Abbreviations: BSA = body surface area, IQR = interquartile range; *n* = number of patients.

**Table 2 cancers-13-03915-t002:** Paclitaxel clearance and Vd of patients with scalp cooling (SC+) versus without scalp cooling (SC−) during one course of paclitaxel administration.

PK Parameter	SC+ (with Scalp Cooling) *n* = 14	SC− (without Scalp Cooling) *n* = 24	SC+ Versus SC−
Clearance *, L/h (CV%)	405.9 (18.2%)	435.5 (21.1%)	-
Relative difference (90% CI)	-	-	−6.8% (−16.7 to 4.4)
Vd *, L (CV%)	234.0 (16.2%)	221.0 (13.0%)	-
Relative difference (90% CI)	-	-	5.9% (−2.3 to 14.8%)

* Clearances and Vd are expressed as geometric means of individual estimates. Abbreviations: CV = coefficient of variation; CI = confidence interval, Vd = volume of distribution.

**Table 3 cancers-13-03915-t003:** Chemotherapy-induced alopecia (CIA) during scalp cooling.

Patient Temperature during SC+	CIA during SC+ CTCAE Grade 1–2 Alopecia *n* = 7	No CIA during SC+ CTCAE Grade 0 Alopecia *n* = 7
Age (years) median [IQR]	50 [47.0–55.5]	59 [46.0–67.5]
Number of courses of paclitaxel administration, median [IQR]	6 [6–18]	6 [6–18]
Paclitaxel dose (mg/m2), median [IQR]	90 [80–90]	90 [80–90]
Paclitaxel dose (mg), median [IQR]	150 [140–170]	150 [130–160]
Concomitant use of carboplatin	5/7	4/7
Ear temperature (°C) baseline versus end of scalp cooling, median [IQR]	36.6 [36.0–36.9] versus 36.2 [34.6–36.6]	36.9 [36.6–37.3] versus 36.6 [35.8–37.5]
Mouth temperature (°C) baseline versus end of scalp cooling, median [IQR]	35.8 [34.8–36.5] versus 35.7 [35.1–36.3]	36.5 [36.3–36.9] versus 36.4 [35.8–37.0]
Axilla temperature (°C) baseline versus end of scalp cooling, median [IQR]	35.6 [35.5–36.1] versus 35.6 [35.2–36.0]	36.0 [35.9–36.5] versus 36.3 [35.7–36.6]
% of patients with a decrease of >1 °C from baseline in at least one of the measurement sites	40%	24%

Abbreviations CIA = chemotherapy-induced alopecia; CTCAE = Common Terminology Criteria for Adverse Events; IQR = interquartile range; SC+ = scalp cooling.

**Table 4 cancers-13-03915-t004:** Paclitaxel clearance and volume of distribution of patients with CIA versus without CIA in the scalp cooling (SC+) subgroup during one course of paclitaxel administration.

PK Parameter	SC+ with CIA *n* = 7	SC+ without CIA *n* = 7	SC+ with CIA Versus SC+ without CIA
Clearance *, L/h (CV%)	410.2 (20.5%)	401.6 (17.1%)	-
Relative difference (95% CI)	-	-	2.2% (−17.8% to 27.1%)
Vd, L (CV%)	234.9 (16.6%)	233.1 (17.2%)	-
Relative difference (95% CI)	-	-	0.2% (−17.1% to 22.5%)

* Clearances and Vd are expressed as geometric means. Abbreviations: CIA = chemotherapy-induced alopecia; CV = coefficient of variation; CI = confidence interval, Vd = volume of distribution.

## Data Availability

The datasets generated during and/or analyzed during the current study are available from the corresponding author upon reasonable request.

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
