# Peer review of "Effect of Scalp Cooling on the Pharmacokinetics of Paclitaxel"

_cancers, 2021, doi:10.3390/cancers13153915_

Round 1

Reviewer 1 Report

In this paper, Van Doom et al. presents results from an original PK study aiming at determining whether scalp cooling does influence the systemic PK of paclitaxel. 

The study is of interest since paclitaxel is frequently used in a variety of cancers frequently encountered in women, and that treatment-related alopecia is a concern in many patients, especially women. 

As expected, scalp cooling does not affect paclitaxel PK - however, this is the first time that such demonstration is made. In this respect,  this study is of interest. I have only a couple of minor remarks.

  1. From a physiological point of view (and a PBPK model perspective), hair represents an epsilon compartment in terms of Vd and Q - the authors should comment on this fact in their Introduction or Discussion.   It probably explains why scalp cooling (i.e., reducing the amount of drugs actually reaching hair only)  is found to have no influence, unlike hypothermia as reported in the Clin PK 2010 review.  Table 3 shows how limited to the scalp is the cooling procedure.
  2.  A graphical representation of the comparative PK profiles of paclitaxel in both groups is critically missing. 
  3. Figures 1 should show SD is reported values are mean values - if it is a representative pâtient this should be underlined. 
  4. Because scalp cooling is all about changing the distribution of the chemo, in addition to clearance values, Vd should be provided as a metrics, especially since PK parameters were estimated using a pop-PK model, making Vd easily available.  Focusing on clearance only is less relevant because (see above) drug disposition (either liver clearance or renal clearance) is unlikely to be affected by scalp cooling just by looking at the reported temperatures of external organs of the upper part of the body in Figure 1. 

Author Response

Response to Reviewer 1.

Comments and Suggestions for Authors

In this paper, Van Doom et al. presents results from an original PK study aiming at determining whether scalp cooling does influence the systemic PK of paclitaxel. The study is of interest since paclitaxel is frequently used in a variety of cancers frequently encountered in women, and that treatment-related alopecia is a concern in many patients, especially women. As expected, scalp cooling does not affect paclitaxel PK - however, this is the first time that such demonstration is made. In this respect, this study is of interest. I have only a couple of minor remarks.

 Response: Thank you very much for these positive comments.  

  1. From a physiological point of view (and a PBPK model perspective), hair represents an epsilon compartment in terms of Vd and Q - the authors should comment on this fact in their Introduction or Discussion. It probably explains why scalp cooling (i.e., reducing the amount of drugs actually reaching hair only) is found to have no influence, unlike hypothermia as reported in the Clin PK 2010 review. Table 3 shows how limited to the scalp is the cooling procedure.

Response: Indeed, deep scalp cooling, lasting for 20-45 min before, during, and lasting for up to 150 minutes after the chemotherapy infusion, leads to a temperature reduction of the whole body. This drop in body temperature may (potentially) lead to alterations in pharmacokinetics. The aim of this analysis therefore was to investigate the effect of scalp cooling on the systemic exposure to paclitaxel. However, the exposure in hair follicles (the epsilon compartment) as a result of scalp cooling was beyond the scope of this analysis. We have made an adjustment in the introduction at line 51-52: ‘This drop in body temperature may lead to ..’ and added reference 7.van den Broek, M.P.; Groenendaal, F.; Egberts, A.C.; Rademaker, C.M. Effects of hypothermia on pharmacokinetics and pharmacodynamics: A systematic review of preclinical and clinical studies. Clin Pharmacokinet 2010, 49, 277-294.

In addition, we have added the following text in the introduction at line 54-59: ‘In a previous study, a physiologically based pharmacokinetic model of doxorubicin was modified to include a scalp skin compartment. The results of the model showed that maximum and average concentration of doxorubicin in the scalp skin compartment were reduced by a factor of 3.6 and 1.6 respectively during scalp cooling. These effects were due to reduced tissue perfusion and can influence the hair follicles survival positively. However, mass transfer characteristics were not considered’ and therefore we added reference 8. Gustafson, D.L.; Rastatter, J.C.; Colombo, T.; Long, M.E. Doxorubicin pharmacokinetics: Macromolecule binding, metabolism, and excretion in the context of a physiologic model. J Pharm Sci 2002, 91, 1488-1501 and 9. Janssen, F.E.M. Modelling physiological and biochemical aspects of scalp cooling. Technische Universiteit Eindhoven 2007.

We have added the following text in the discussion at line 266-272 ‘In our analysis, a population PK model consisting of a central compartment with two peripheral compartments connecting to it was used to obtain clearances and volumes of distribution of each subject. Unlike PBPK models, these compartments have no anatomic or physiological significance. However, it can still be used to investigate the influence of subject characteristics (ie. scalp cooling) on the predicted subject PK parameters describing the whole-body drug disposition. Considering the scalp skin drug disposition in particular would require more data and it is outside the scope of our study’ and added reference 23 Mould, D.R.; Upton, R.N. Basic concepts in population modeling, simulation, and model-based drug development. CPT Pharmacometrics Syst Pharmacol 2012, 1, e6.

2.  A graphical representation of the comparative PK profiles of paclitaxel in both groups is critically missing. 

Response: We agree with the reviewer, and therefore we have added a graphical representation of the PK profiles of paclitaxel as the new Figure 1. with legends to this figure (see Line 198-205).

 (Viewable in attached coverletter, and in the revised manuscript)

Figure 1: Paclitaxel concentrations (on log-transformed data) of the pharmacokinetic profile of patients using scalp cooling (n=14, blue line) and patients not using scalp cooling (n=24, red line).

3. Figure 1 should show SD if reported values are mean values - if it is a representative patient this should be underlined. 

Response: As a result of the previous comment by this reviewer, the original Figure 1 was changed in Figure 2, which shows the median body temperature of all 14 patients. For clarity, we added this to the figure legend of Figure 2.

4. Because scalp cooling is all about changing the distribution of the chemo, in addition to clearance values, Vd should be provided as a metrics, especially since PK parameters were estimated using a pop-PK model, making Vd easily available.  Focusing on clearance only is less relevant because (see above) drug disposition (either liver clearance or renal clearance) is unlikely to be affected by scalp cooling just by looking at the reported temperatures of external organs of the upper part of the body in Figure 1. 

Response: We have followed the reviewer’s suggestion and we added the Vd estimates. In our analysis, a population PK model consisting of a central compartment with two peripheral compartments connecting to it was used to obtain clearances and volumes of distribution of each subject. Unlike PBPK models, these compartments have no anatomic or physiological significance. However, it can still be used to investigate the influence of subject characteristics (i.e. scalp cooling) on the predicted subject PK parameters describing the whole-body drug disposition. Considering the scalp skin drug disposition in particular this would require more data which was outside the scope of our study. 

Reviewer 2 Report

In their manuscript, van Doorn and colleagues evaluated the effects of scalp cooling on the pharmacokinetics of paclitaxel in female patients with cancer.

The purpose of the study is clear and of interest. I appreciated the opportunity to review this paper. The manuscript is overall well written, although I would advise for some proof-reading (e.g., check lines 50, 88, 100, 103-104, 114, 116, 117, 230).

I have no major concerns regarding the statistical analysis. I think the paper will be a good contribution to the literature.

In the Discussion (line 230), the authors mentioned that the drop observed in temperature in this study was maybe too small to demonstrate a difference in clearance of paclitaxel. Can the authors elaborate on that? Was the drop in temperature smaller than expected or smaller compared to other studies using scalp cooling? Should future studies monitor the body temperature more frequently to get a better overview of the temperature dynamics?

Author Response

Responses to Reviewer 2.

Comments and Suggestions for Authors

In their manuscript, van Doorn and colleagues evaluated the effects of scalp cooling on the pharmacokinetics of paclitaxel in female patients with cancer.

The purpose of the study is clear and of interest. I appreciated the opportunity to review this paper. The manuscript is overall well written, although I would advise for some proof-reading (e.g., check lines 50, 88, 100, 103-104, 114, 116, 117, 230).

Response: Thank you very much for your positive comments. We have followed your advice and made adjustments to the indicated lines after the revisited manuscript lines 50 (addition lead), 88 (addition,), 103 (change axilla in axilas),110 (addition (were), 114 addition (change administrated in administered), 125 addition (was conducted), 127 change (lineare limination in linear limitation), 282 (230) addition (remove this) and we have done proof-reading of the whole document.

I have no major concerns regarding the statistical analysis. I think the paper will be a good contribution to the literature.

Response: Thank you very much for these comments.

In the Discussion (line 230), the authors mentioned that the drop observed in temperature in this study was maybe too small to demonstrate a difference in clearance of paclitaxel. Can the authors elaborate on that? Was the drop in temperature smaller than expected or smaller compared to other studies using scalp cooling? Should future studies monitor the body temperature more frequently to get a better overview of the temperature dynamics?

Response: Maybe we caused some confusion with our remark. What we meant to say is that as a result of the cold cap the body temperature just slightly decreased. We did not expect a major change on forehand, so that was not surprising to us. It is also in line with other work. We don’t feel that future studies should monitor body temperature, as we have no reason to suspect an influence of the pharmacokinetics of paclitaxel.

Reviewer 3 Report

Although having some interesting hint the work is not quite detailed, it doesn't sound good and do not deserves the publication in Cancer.

I believe that the number of patients enrolled in the study is too limited to draw firm conclusions even if the statistical approach can be considered correct

The experimental data come from a single experiment, I believe that the samples should be repeated on the various subjects in subsequent weeks to give greater reliability to the paper.

Author Response

Responses to Reviewer 3.

Comments and Suggestions for Authors

Although having some interesting hint the work is not quite detailed, it doesn't sound good and do not deserves the publication in Cancer.

Response: This is unfortunate to read of course. However, we feel that our answers to the specific comments of this reviewer will be helpful and convince for this reviewer that our work is interesting enough for publication in Cancers.

I believe that the number of patients enrolled in the study is too limited to draw firm conclusions even if the statistical approach can be considered correct.

Response: We do not completely understand the reasoning of the reviewer. In the design of this study proper statistical methods were used in close collaboration with our statistician. The design was also judged by the Medical Ethical Review Board and was found appropriate for our research question without any concerns about the sample size. Also, in previous work we used relatively small sample sizes (i.e. van Leeuwen et al, J Clin Oncol 2016, de Man et al, Clin Pharmacol Ther 2019, Atrafi et al, Clin Cancer Res 2020, Veerman et al, Clin Pharmacokinet 2021, etc.). Yet, the designs are solid and papers are published in highly ranked peer reviewed journals. Nevertheless, we understand that at first sight the number of patients seem limited to readers who are not used to this type of design. However, the question is what the reviewer would expect if the sample size would have been larger? In general, larger sample sizes will narrow the confidence intervals around the studied effects which would make our conclusions even stronger if we assume that the point estimates would not change much by the addition of more patients. Hence, we are confident that we can draw solid conclusions despite the relatively small number of patients. We are convinced that scalp cooling does not influence the pharmacokinetics of paclitaxel and is safe to be used in clinical practice. This message is of clinical relevance and therefore we feel the reader should be informed about our findings.

The experimental data come from a single experiment I believe that the samples should be repeated on the various subjects in subsequent weeks to give greater reliability to the paper.

Response: As replied to the reviewer in our answer to the previous question, the design is solid enough, also from a statistical point of view. We agree with the reviewer that a design in which patients were exposed to the cold cap in one course and no cold cap in the next one would be scientifically sound. However, this would be unethical for the cancer patients, as they would like to keep their hair, and a course without cold cap would result in alopecia. Our Ethical Board would never have approved such a design. Therefore, we have chosen for this design to find a good balance between our research questions and the burden of the patient.

Round 2

Reviewer 3 Report

The few changes made add value to the work which is now worthy of publication